# Reporting quality and spin in abstracts of randomized clinical trials of periodontal therapy and cardiovascular disease outcomes

Murad Shaqman[1], Khadijeh Al-Abedalla[2], Julie Wagner[3], Helen Swede[4], John Cart Gunsolley[5], Effie Ioannidou[2]*

1 Department of Oral Maxillofacial Surgery, Oral Medicine and Periodontology, School of Dentistry, The University of Jordan, Amman, Jordan, 2 Division of Periodontology, School of Dental Medicine, UCONN Health, Farmington, CT, United States of America, 3 Division of Behavioral Science, School of Dental Medicine, UCONN Health, Farmington, CT, United States of America, 4 Department of Community Medicine, School of Medicine, UCONN Health, Farmington, CT, United States of America, 5 Department of Periodontics, School of Dentistry, Virginia Commonwealth University, Richmond, VA, United States of America

☯ These authors contributed equally to this work.

* Ioannidou@uchc.edu

**Data Availability Statement:** All relevant data are within the paper and its Supporting Information files.

## Abstract

### Objective

Poor reporting in randomized clinical trial (RCT) abstracts reduces quality and misinforms readers. Spin, a biased presentation of findings, could frequently mislead clinicians to accept a clinical intervention despite non-significant primary outcome. Therefore, good reporting practices and absence of spin enhances research quality. We aim to assess the reporting quality and spin in abstracts of RCTs evaluating the effect of periodontal therapy on cardiovascular (CVD) outcomes.

### Methods

PubMed, Scopus, the Cochrane Central Register of Controlled Trials (CENTRAL), and 17 trial registration platforms were searched. Cohort, non-randomized, non-English studies, and pediatric studies were excluded. RCT abstracts were reviewed by 2 authors using the CONSORT for abstracts and spin checklists for data extraction. Cohen's Kappa statistic was used to assess inter-rater agreement. Data on the selected RCT publication metrics were collected. Descriptive analysis was performed with non-parametric methods. Correlation analysis between quality, spin and bibliometric parameters was conducted.

### Results

24 RCTs were selected for CONSORT analysis and 14 fulfilled the criteria for spin analysis. Several important RCT elements per CONSORT were neglected in the abstract including description of the study population (100%), explicitly stated primary outcome (87%), methods of randomization and blinding (100%), trial registration (87%). No RCT examined true outcomes (CVD events). A significant fraction of the abstracts appeared with at least one form of spin in the results and conclusions (86%) and claimed some treatment benefit in

**Funding:** This manuscript received NIH/NIDDK funding (R21 DK108076) awarded to EI.

**Competing interests:** No authors have competing interest.

spite of non-significant primary outcome (64%). High-quality reporting had a significant positive correlation with reporting of trial registration ($p = 0.04$) and funding ($p = 0.009$). Spinning showed marginal negative correlation with reporting quality ($p = 0.059$).

## Conclusion

Poor adherence to the CONSORT guidelines and high levels of data spin were found in abstracts of RCTs exploring the effects of periodontal therapy on CVD outcomes. Our findings indicate that journal editors and reviewers should consider strict adherence to proper reporting guidelines to improve reporting quality and reduce waste.

## Introduction

The abstract of randomized clinical trials (RCT) provides the reader with the first account of the trial objectives, methodology and results. Therefore, reporting accuracy, clarity and quality have a critical role during the initial assessment of the trial and affects the decision to read the full text [1]. Furthermore, in many geographic locations, RCT abstracts are often the only section of an RCT freely accessible to clinicians [2].

In recognition of the importance of RCT abstracts, the Consolidated Standards of Reporting Trials (CONSORT) for abstracts guidelines [3] were developed as an extension to the original CONSORT, addressing clarity, completeness and transparency and ensuring that key trial elements are properly reported. Hence, poor reporting refers to omitting important information in abstracts as required by the well-defined CONSORT items [2].

Furthermore, spin is defined as failure to accurately and faithfully report the findings of a scientific study in a manner that would affect the reader's perception of the outcomes [4]. The tool for spin assessment in publication abstracts [4] identifies reporting practices that constitute an intentional or unintentional attempt to spin the results and/or conclusions leading to misreporting and bias. Despite the development of reporting and spin guidelines, abstracts in biomedical literature are often characterized by poor reporting quality and biased finding interpretation [5–12].

The impact of poor reporting and spin on the public and professional perception of research findings is discernible. In fact, abstracts with high levels of spin were found to be more frequently read compared to abstracts of the same trial after being edited to omit spin, and were also more likely to mislead clinicians to accept a clinical intervention as being beneficial despite a non-significant primary outcome [1]. Moreover, spin in abstracts percolates into media coverage and press releases, which in turn generates greater public attention [13] Paradoxically, articles that received greater media attention showed improved citation metrics in subsequent publications [14], creating what resembles of a vicious circle of public and scientific misinformation.

Ever since the publication of the earliest studies indicating a correlation between cardiovascular (CVD) disease and periodontitis [15, 16], the findings have received considerable professional and public interest. To test causal relationships, several RCTs explored the effect of periodontal therapy on CVD outcomes. Subsequently, the topic sparked intense debates between researchers, caused wide-scale media coverage and public interest, and prompted involved professional organizations to issue official statements [17, 18].

Although multiple periodontal-CVD RCTs have been published, the adherence to the CONSORT guidelines and the incidence of spin has not been studied. Therefore, the aim of

this study was to evaluate the reporting quality and the incidence of spin in abstracts of RCTs investigating the effect of periodontal therapy on CVD disease outcomes.

# Materials and methods

## Search methods and study selection

Studies were retrieved from PubMed, Scopus based on search strategy shown below. In addition, we crosschecked 17 trial registration platforms included in the World Health Organization International Clinical Trials Registry Platform [19] to confirm trial registration status and information (S1 Table). The search was conducted for all registers on 01/01/2018.

Search keywords and limitations or filters for each database were as follows:

a. Pubmed: ("Lipids"[Mesh] OR "Acute-Phase Proteins"[Mesh] OR "Blood Pressure"[Mesh] OR "Arterial Pressure"[Mesh] OR "Hypertension"[Mesh] OR "Hypotension"[Mesh] OR "Cholesterol"[Mesh] OR "Cholesterol, LDL"[Mesh] OR "Cholesterol, HDL"[Mesh] OR "Cholesterol Esters"[Mesh] OR "Embolism, Cholesterol"[Mesh] OR "Cholesterol, VLDL"[Mesh] OR "Cardiovascular System"[Mesh] OR "Cardiovascular Infections"[Mesh] OR "Cardiovascular Abnormalities"[Mesh] OR "Cardiovascular Diseases"[Mesh] OR "Cardiovascular Physiological Phenomena"[Mesh] OR "Endothelium"[Mesh] OR "Endothelial Cells"[Mesh]) AND ("Periodontal Debridement"[Mesh] OR "Periodontal Diseases"[Mesh] OR "Periodontal Pocket"[Mesh] OR "Alveolar Bone Loss"[Mesh] OR "Dental Scaling"[Mesh] OR "Periodontitis"[Mesh] OR "Dental Prophylaxis"[Mesh] OR "Periodontal Attachment Loss"[Mesh]) filter: clinical trial

b. Scopus: TITLE-ABS-KEY (("Lipids" OR " Proteins*" OR "Pressure*" OR "Hypertension" OR "Hypotension" OR "Cholesterol" OR "Cardiovascular*" OR "Endothelium" OR "Endothelial*") AND ("Periodontal*" OR "Alveolar Bone Loss" OR "Dental*" OR "Periodontitis") AND "clinical trial") AND (LIMIT-TO (DOCTYPE, "ar"))

c. 17 trial registration platforms (S1 Table) were searched. Since these platforms were limited to one or two keywords, "periodont*" was used as a main keyword, then records were scanned for eligible studies.

The retrieved articles were hand-screened for identification of additional RCT reports (Fig 1), and then duplicates were excluded.

## RCT report inclusion criteria

1. Study Design:
   Only publications of periodontal-CVD RCTs were included. Cohort, non-randomized trials or observational trials were excluded. RCT publications in languages other than English were excluded.

2. Participants:
   Targeted populations included adult patients with no systemic diseases other than CVD diseases. Studies were included if the participants were diagnosed with chronic periodontitis only. Studies with participants diagnosed with aggressive periodontitis, gingivitis, or peri-implantitis were excluded.

3. Intervention:
   The tested intervention included subgingival scaling and root planing (SRP) or SRP with

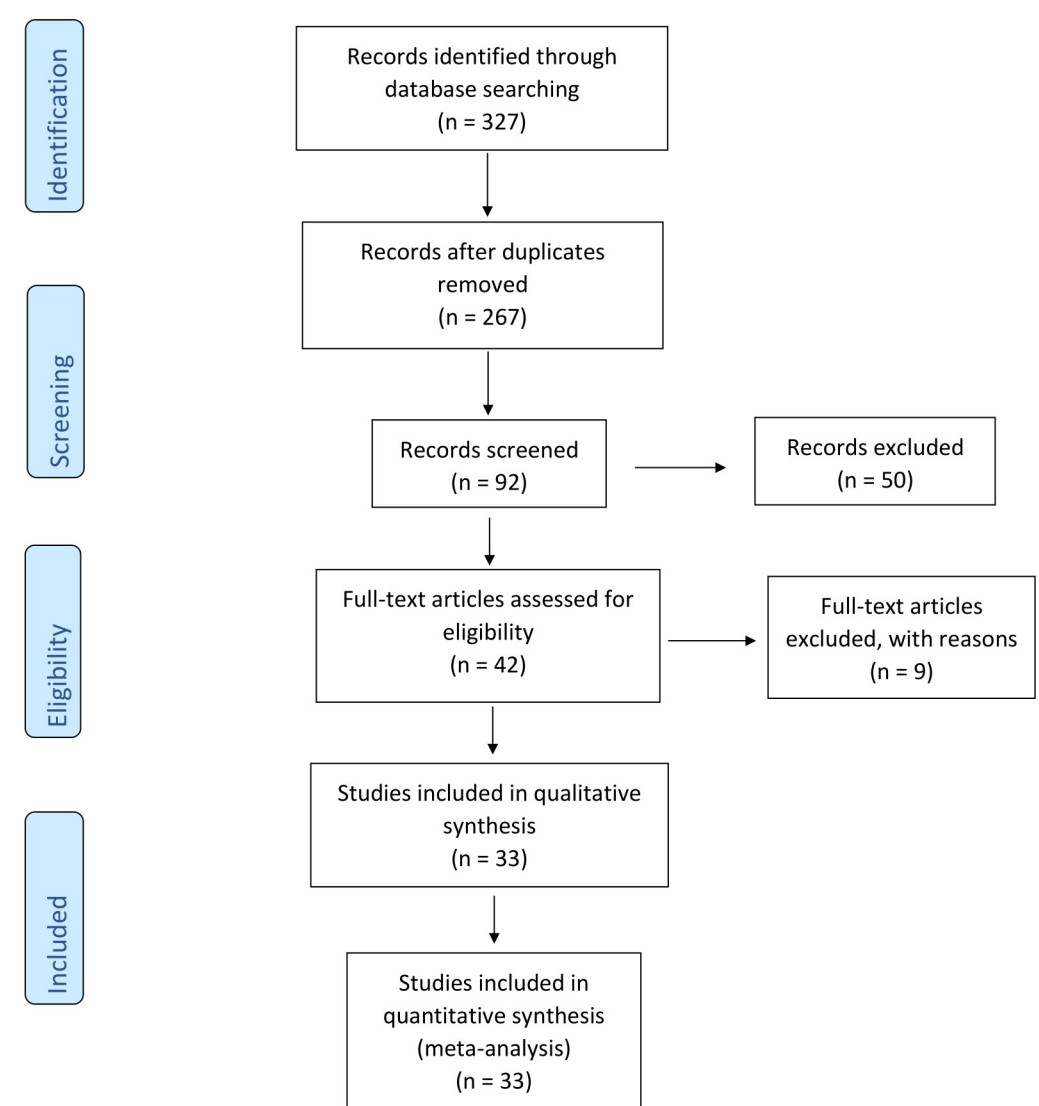

**Fig 1. Search Strategy with flow chart representing the identification, screening, eligibility and inclusion stages.**

adjunctive therapy. Interventional studies employing adjuncts alone, supragingival scaling alone or surgical therapy were excluded.

4. Outcomes:

   True or surrogate CVD outcomes were included. For descriptive purposes, outcomes were segregated into two groups [20]:

   a. CVD true events; such as angina, myocardial infarction, stroke, and CVD end points (CVD related-death).

   b. Surrogate outcomes; such as blood pressure, lipids, blood tests, ugh sensitivity C-reactive protein (hs-CRP), lipoproteins, and blood cell count.

**Additional selection criteria for spin assessment.** Only studies with a clearly defined primary outcome were included in the spin analysis. To fulfill this condition, the primary outcome should be either explicitly stated in the abstract or the full text. In cases were the primary

outcome was not explicitly reported, the outcome stated in the sample size calculations was considered as the primary outcome. If no outcome was stated in the sample size calculation, we deduced a primary outcome based on the stated objectives of the study. If no primary outcome could be identified, the study was excluded. In addition, studies with multiple primary outcomes were excluded.

## Data extraction and compilation

1. Selection of studies was carried out according to the inclusion criteria. Titles and abstracts of the search results were initially screened for identification of other potential eligible studies. Then, full texts were retrieved and assessed to further assess eligibility. In cases of multiple published reports associated with the same trial registration number, the primary publication on the RCT results was selected to avoid duplication.

2. Data extraction: Two authors (MS, KA) independently reviewed the abstracts -and the full texts, when needed- of the included RCT reports, and applied the CONSORT for abstracts [3] and the SPIN checklist [1]. Disagreements were resolved by a third author (EI). Two items of the CONSORT abstracts guidelines were excluded because they only apply to unpublished studies or conference abstracts.

3. Characteristics of each RCT abstract and the respective publishing journal were extracted:

   a. Abstract word count

   b. Number of citations as shown on Scopus [21].

   c. Trial registration number, trial registration date was determined based on the information provided by the trial registry.

   d. Trial funding source, number of authors, geographic location,

   e. Trial sample size, intervention and outcomes.

   f. Journal metrics such as 5 years-impact factor, impact factor without self-citation, influence factor as reported on Thomson-Reuters/Clarivate Analytics 2018 [22].

4. A decision-making guide was implemented to assist with the calibration and review process [23]. An Overall CONSORT Score for reporting quality and Overall Spin Score were calculated for each RCT publication based on the CONSORT and Spin checklist.

## Statistical analysis

Cohen's Kappa statistic was used to assess inter-rater agreement. For the descriptive analysis categorical variables were expressed as proportion percent. For the exploratory bivariate association between Overall CONSORT Score and related variables, we applied a Spearman correlation model. The limited sample size did not allow for a further multivariate regression model to assess predictors of reporting quality. Statistical analysis was performed with SPSS software (SPSS Inc).

## Results

### General findings

Twenty-four RCT reports were deemed eligible and entered the analysis (Fig 1). Among them, one study was a secondary analysis publication [24]. The PAVE study had multiple publications, and according to our inclusion criteria we included only the results publication [25]

**Table 1. Characteristics of included publications and publishing journals.**

| RCT | Year | Journal metrics | | | | | Article metrics | | | | | |
|---|---|---|---|---|---|---|---|---|---|---|---|---|
| | | Journal | 5 years impact | Impact factor without self-citation | Eigenfactor | Influence factor | Abstract Word count | Authors # | Citations | Registration | Geographic location | Funding |
| [24] | 2017 | JCP | 4.62 | 3.34 | 0.011 | 1.15 | 208 | 5 | 3 | Yes | Netherlands | Yes |
| [25] | 2009 | JP | 3.52 | 3.02 | 0.011 | 0.86 | 241 | 17 | 13 | Yes | USA | Yes |
| [26] | 2012 | JCP | 4.62 | 3.34 | 0.011 | 1.15 | 193 | 7 | 20 | Yes | Pakistan | Yes |
| [27] | 2015 | Medicine | 2.19 | 1.89 | 0.05 | 0.59 | 237 | 11 | 132 | No | Italy | No |
| [28] | 2014 | JCP | 4.62 | 3.34 | 0.011 | 1.15 | 174 | 4 | 198 | No | Brazil | Yes |
| [29] | 2015 | JP | 3.52 | 3.02 | 0.011 | 0.86 | 266 | 5 | 113 | Yes | Brazil | Yes |
| [30] | 2005 | J Dent Res | 5.72 | 5.07 | 0.02 | 1.55 | 160 | 5 | 40 | No | UK | Yes |
| [31] | 2006 | Am Heart J | 4.63 | 4.08 | 0.04 | 2.11 | 346 | 6 | 27 | No | UK | Yes |
| [32] | 2016 | Clin Oral Invest | 2.55 | 2.23 | 0.011 | 0.68 | 240 | 5 | 682 | No | China | Yes |
| [33] | 2007 | S Med J | 0.96 | 0.81 | 0.002 | 0.32 | 145 | 7 | 36 | No | Turkey | No |
| [34] | 2011 | JP | 3.52 | 3.02 | 0.011 | 0.86 | 239 | 3 | 12 | No | India | No |
| [35] | 2015 | JP | 3.52 | 3.02 | 0.011 | 0.86 | 277 | 4 | 1 | Yes | India | No |
| [36] | 2003 | JCP | 4.62 | 3.34 | 0.011 | 1.15 | 174 | 6 | 27 | No | UK | Yes |
| [37] | 2016 | Lasers Surg Med | 2.74 | 2.49 | 0.004 | 0.59 | 249 | 8 | 16 | No | Saudi Arabia | Yes |
| [38] | 2011 | J Perio Res | 2.71 | 2.70 | 0.004 | 0.63 | 253 | 5 | 1 | No | Jordan | Yes |
| [39] | 2014 | Hypertension | 6.74 | 6.16 | 0.04 | 2.19 | 254 | 13 | 40 | Yes | Australia | Yes |
| [40] | 2011 | JCP | 4.62 | 3.34 | 0.011 | 1.15 | 192 | 5 | 1 | Yes | China | Yes |
| [41] | 2012 | JP | 3.52 | 3.02 | 0.011 | 0.86 | 310 | 6 | 88 | No | Chile | Yes |
| [42] | 2010 | Eur J Oral Sci | 1.91 | 1.57 | 0.003 | 0.56 | 177 | 11 | 24 | No | Australia | Yes |
| [43] | 2007 | NEJM | 67.51 | 78.54 | 0.70 | 29.45 | 255 | 10 | 8 | No | UK | Yes |
| [44] | 2007 | JCP | 4.62 | 3.34 | 0.011 | 1.15 | 201 | 12 | 240 | No | Turkey | Yes |
| [45] | 2008 | JCP | 4.62 | 3.34 | 0.011 | 1.15 | 189 | 9 | 16 | No | Japan | Yes |
| [46] | 2009 | JP | 3.52 | 3.02 | 0.011 | 0.86 | 228 | 4 | 5 | No | Brazil | Yes |
| [47] | 2017 | JP | 3.52 | 3.02 | 0.011 | 0.86 | 252 | 10 | 25 | Yes | China | Yes |

JCP: Journal of Clinical Periodontology, JP: Journal of Periodontology, J Dent Res: Journal of Dental Research, Am Heart J: American Heart Journal, Clin Oral Invest: Clinical Oral Investigations, J Perio Res: Journal of Periodontal Research, Eur J Oral Sci: European Journal Oral Science, S Med J: Southern Medical Journal, Lasers: Surg Med: Lasers in Surgery and Medicine, NEJM: New England Journal of Medicine.

For each trial, journal and article metrics are presented in Table 1.

An overview of other outcomes of the included articles is shown Table 2.

Generally, all RCTs explored surrogate outcomes and none examined CVD events (Table 2). Only 3 abstracts had explicitly stated primary outcome, while for the remaining RCTs, we identified and extracted the primary outcome from the full text (Table 2). RCTs with more than one outcome were excluded from spin analysis, as primary outcome identification was impossible.

In terms of trial interventions, Table 2 shows that SRP alone was used in 58.3% of the included RCTs, while SRP and adjuncts were used in 41.7%. The majority of the RCTs reported some type of funding (83.3%). Specifically, 58% were funded intramurally, 46% by foundations, 29% by federal agencies, and 25% by industry. 60% of the funded RCTs reported multiple funding sources.

Abstract word count for each trial is presented in Table 1, and was categorized into 3 groups according to CONSORT findings [3], which is presented in Table 2. 66.7% of the abstracts had <250-word count, 25% had word count from 250–300, and 8.3% had word count >300 [3].

**Table 2. Descriptive data on primary outcome and interventions.**

| | CHARACTERISTIC | N = 24 | % |
|---|---|---|---|
| 1 | Nature of primary outcome | | |
| | True outcome (CV event) | 0 | 0 |
| | Surrogate outcome | 24 | 100 |
| 2 | Primary outcome source | | |
| | Explicitly stated in abstracts | 3 | 13% |
| | Explicitly stated in full texts | 3 | 13% |
| | Based on power analysis | 7 | 29% |
| | Implied in objectives | 11* | 33% |
| 3 | Intervention type | | |
| | SRP alone | 14 | 58% |
| | SRP + adjunct | 10 | 42% |

\* the outcomes of 11 papers were implied by objectives and were more than one primary outcome, they weren't included in the spin analysis

## CONSORT checklist findings

Following the RCT abstract assessment, Overall CONSORT Score ranged from 2 to 9 out of 15. Table 3 presents the frequency of each CONSORT item fulfillment.

Specifically, only 50% of the included RCTs acknowledged the term "randomized clinical trial" in the title. Only 3 of the studies (13%) included the specific study design in the abstract (i.e. parallel group, crossover, superiority, etc.).

**Assessment of methods reporting.** Three items in the methods section lacked reporting in all RCTs, including the item "participants", which lacked information about the location of the study and the detailed description of the participants that were included, and the item "randomization", were all of the studies did not report the randomization method that was used, and the item "blinding", were studies did not report the level blinding. The item

**Table 3. Fulfillment (%) of CONSORT items stratified by CONSORT sections.**

| CONSORT FOR ABSTRACT CHECK LIST | (NUMBER) PERCENTAGE |
|---|---|
| **TITLE** | 12 (50%) |
| **TRIAL DESIGN** | 3 (13%) |
| **METHODS** | |
| PARTICIPANTS | 0.0% |
| INTERVENTIONS | 13 (54%) |
| OBJECTIVE | 22 (92%) |
| OUTCOME | 3 (13%) |
| RANDOMIZATION (METHOD) | 0 (0%) |
| BLINDING (MASKING) | 0 (0%) |
| **RESULTS** | |
| NUMBERS RANDOMIZED | 20 (83%) |
| NUMBERS ANALYZED | 4 (17%) |
| OUTCOME | 3 (13%) |
| HARMS | 1 (4%) |
| **CONCLUSIONS** | 3 (13%) |
| **TRIAL REGISTRATION** | 3 (13%) |
| **FUNDING** | 8 (33%) |

"intervention" mostly lacked the necessary detailed description of the intervention; therefore, only 54% of the RCTs fulfilled this item.

Most of the study abstracts (92%) included the objectives of the study. Interestingly, only 3 abstracts (18.5%) explicitly stated the primary outcome.

**Assessment of results reporting.**   Although 83% of the abstracts included the numbers of randomized populations, only 17% included the numbers of analyzed populations as part of the abstract materials and methods rather than the results section violating the CONSORT recommendations. Only 1 RCT indicated the harms in the abstracts (4%).

**Assessment of conclusion, registration and funding reporting.**   Only 13% discussed the results and conclusion of the primary outcome. Trial registration information was reported in only 3 abstracts (13%). When the public trial registration records were examined using the registration number included in the study, we found that 4 RCTs registered following the study initation and the first subject recruitment.

## Spin analysis findings

After applying the exclusion criteria as outlined in the methodology, 14 out of the 26 RCT reports were included in the spin analysis (S2 Fig). The prevalence and type of spin for the included articles is outlined in Table 4.

**Table 4.  Fulfillment (%) of SPIN items in result and conclusion sections.**

| Type of spin | (number) Percentage |
|---|---|
| 1) spin in the result | |
| Focus on statistically significant within-group comparison | 3 (21%) |
| Focus on statistically significant secondary outcomes | 9 (64%) |
| Focus on statistically significant subgroup analyses | 2 (14%) |
| Focus on statistically significant modified population of analyses (eg, per-protocol analyses) | 4 (29%) |
| Focus on statistically significant within- and between-group comparisons for secondary outcomes | 9 (64%) |
| OTHER SPIN: NO DEFINITION OF PRIMARY OR SECONDARY OUTCOMES | 11 (79%) |
| 2) spin in the conclusions | |
| Focus only on treatment effectiveness: | |
| 1. Claiming equivalence for statistically nonsignificant results | 0 (0%) |
| 2. Claiming efficacy with no consideration of the statistically nonsignificant primary outcome | 9 (64%) |
| 3. Focusing only on statistically significant results | 6 (43%) |
| Acknowledge statistically nonsignificant results for the primary outcome but emphasize the beneficial effect of treatment | 9 (64%) |
| Acknowledge statistically nonsignificant results for the primary outcome but emphasize other statistically significant results | 9 (64%) |
| Other spin in Conclusions section: | |
| 1. Conclusion ruling out an adverse event on statistically nonsignificant results | 0 (0%) |
| 2. Conclusion focusing on within-group assessment (both treatments are effective/treatment administered in both groups is effective (eg, add-on studies) | 2 (14%) |
| 3. Recommendation to use the treatment | 1 (7%) |
| 4. Focus on another objective | 1 (7%) |
| 5. Comparison with placebo group of another trial | 0 (0%) |
| 6. Statistically nonsignificant subgroup results reported as beneficial | 0 (0%) |
| Others: Inadequate extrapolation to larger population, intervention or outcome | 12 (86%) |
| misleading statements designed to exaggerate or falsely claim efficacy | 7 (50%) |
| 3) Spin in both results and conclusions | 12 (86%) |

Some form of spin in both of the results and conclusions sections was detected in the majority of the RCTs (86%). Given that 79% of the included studies failed indicate the primary or secondary outcomes in the abstracts, we considered that these studies employed diverse strategies of spin.

In the results section, at least one checklist item showed a form of spin. 64% of the studies focused on statistically significant secondary outcomes, and on statistically significant within- and between- group comparisons of secondary outcomes.

In the conclusion sections, half of the included RCTs (50%) made statements that were misleading and designed to exaggerate or falsely claim efficacy. 43% of the conclusions were focusing only on significant results regardless if they corresponded to the primary outcome, 7% focusing on another objective, and 7% making treatment recommendations. 64% acknowledged statistically non-significant results for the primary outcome yet emphasized the beneficial effect of treatment, and emphasized other statistically significant results.

## Bivariate correlation analyses

The Overall CONSORT Score ranged between 2–9 out of 15 with some articles fulfilling only had 2 items of the CONSORT checklist. The maximum number of fulfilled items by a single publication was 9 out of 15 (S1 Fig). The Overall Spin Score ranged between 1–13. Some publications included only 1 item with some form of spin, while some articles had 13 items that were spun (S2 Fig).

Within the limitations of the study, there was a positive and significant bivariate correlation between the Overall CONSORT Score and funding source (correlation coefficient: 0.416, P-value: 0.043). In addition, we observed a significant correlation between Overall CONSORT Score and registration reporting in the abstract (correlation coefficient: 0.518, P-value: 0.009). In summary, we observed that abstracts that included trial registration and funding information were characterized by higher reporting quality.

Overall Spin Score showed a marginal negative correlation with Overall CONSORT Score (correlation coefficient: -0.517, P-value: 0.059), which signified that the higher the reporting quality in the abstract, the lower the spin. Overall Spin Score showed negative correlation, although not significant, with funding and registration. Overall Spin Score showed marginal negative correlation with the number of publication authors (correlation coefficient of -0.509, P-value of 0.063).

## Discussion

Our study evaluated the reporting quality and incidence of spin in the abstracts of 24 RCT publications assessing the impact of periodontal interventions on CVD outcomes. To our knowledge, this is the first paper that evaluated both reporting quality and spin in abstracts of such publications. The overall reporting quality of the included abstracts was deemed to be poor. Overall, we found that the RCT objectives and numbers of randomized subjects were the only most adequately reported items (92% and 83% respectively). All other CONSORT items were adequately reported in less than 50% of the abstracts. Notably, we found limited RCT abstracts with adequate reporting on the exact trial design (17%), method of randomization (0%), blinding (0%), number of subjects analyzed (17%), harms (4%), outcomes in both trial arms (13%), as well as the interpretation of the results in the conclusions (13%). Our findings were in agreement with other studies in the medical and dental literature confirming inadequate reporting according to CONSORT guidelines [9, 48–50]. Surprisingly, even after dental journals adhered to the CONSORT for abstracts guidelines [50], those guidelines were not

systematically reinforced. Therefore, RCT abstracts were still characterized by inadequate reporting quality.

Consistent with other reports, the CONSORT items most adequately reported in the RCT abstracts were related to objectives and numbers randomized [49, 50]. Journal and article metrics including impact factor or citation metrics were unreliable in predicting reporting quality, as confirmed in other studies [51]. The lack of significance in this correlation could be also related to the small number of included publications.

In the spin analysis, it is noteworthy, that 10 of 24 of the included RCT publications were excluded due to the lack of an explicitly defined single primary outcome. The use of multiple primary outcomes in RCTs combined with the lack of adequate power analysis for multiple outcomes might significantly increase the risk of bias.

The spin analysis according to established criteria [4] showed that various strategies of spin were adapted in the included abstracts (n = 14). Specifically, spin phenomena in either the result or conclusion sections of the abstracts were detected in the majority of the publications. Half of the abstracts presented a tendency for conclusions that had misleading statements designed to exaggerate or falsely claim efficacy. One third of the RCT abstracts presented the trial results in a before-after therapy manner focusing on within group analysis, highlighting statistical significance and ignoring the between group comparisons as directed by the study objectives. More than half of the RCT abstracts emphasized significance in secondary outcomes, a commonly used spin strategy, when the primary outcome results were not significant.

Our results agreed with other studies in the medical literature that investigated spin strategies and misrepresentation of RCT results with various methodologies [4, 5, 51–54]. Austin et al [5] reported some form of spin to exist in 47% of the included RCT abstracts while Cooper et al [53] reported spin to be as high as 70% of the included articles. Interestingly, Pitkin et al [8] compared data reported in the abstract of a random sample of RCTs published in 6 major medical journals to the data presented in the full-text manuscripts and found that inconsistencies at variable levels (18%-68%) existed between data reported in the abstract compared to the full text.

The present study has several strengths. We applied strict inclusion criteria as directed by the research hypothesis and only included RCT publications examining the impact of periodontal intervention on CVD outcomes [55]. We standardized the data extraction methodology utilizing well-defined decision guide and calibration between assessors. Therefore, we have demonstrated a high level of inter-rater agreement with any differences resolved by a third evaluator to ensure greatest accuracy in our analysis. While the present study focused on RCT abstracts alone and not the full text of the included publications, these considerable reporting shortages and/or misrepresentations were a cause for concern, given the wide attention abstracts receive within the healthcare and media communities.

Our study also has some inherent limitations. Although the spin assessment is characterized by subjectivity, two independent and calibrated reviewers per abstract conducted the data extraction and determined the spin strategies. With this method, we aimed to control the magnitude of subjectivity. We employed spin analysis previously used by other groups [1, 4, 6, 53]. Therefore, our analysis was focused on abstract sections and might have missed additional spin strategies present in the full text.

It is important to emphasize that poor reporting quality does not translate into poor study design [56]. It does, however, indicate lack of transparency and prevent the replication of the given experiment [57]. Therefore, quality reporting is necessary for the advancement of science [58]. It is also even more important to emphasize that the identification of spin in an abstract (according to criteria of Boutron et al 2014) should not be taken as a verdict that a

research report is fraudulent or fake. The spin strategies examined don't all carry the same weight in terms of the impact on a reader's perception of the abstract and it does not assess if these spin strategies were applied in the full text of the manuscript. Nevertheless, low quality reporting and introduction of spin might be contributing to continued controversy in this field of research [59], flawed professional and public perception of research findings [1, 13] and continued ill-advised expenditure of valuable time and resources [60]. The responsibility to improve reporting of RCTs and avoidance of misrepresentations falls on multiple parties. Journal editors and peer reviewers as gatekeepers could reinforce strict practices to ensure adherence to CONSORT or other reporting guidelines, and to require trial registration prior to the commencement of the trial as recommended by the ICMJE [61]. An additional effort by academic institutions, professional organizations, and scientific communities should be exerted to raise awareness among the general scientific audience on proper reporting practices and spin strategies. The scientific community should embrace post-publication appraisal and critique with a goal to improve reporting quality and minimize the incidence of spin.

Conclusions: Poor adherence to the CONSORT for abstracts guidelines and high levels of data "spin" were found in the abstracts of RCTs examining the effect of periodontal therapy on CVD outcomes. Our findings indicate that journal editors and reviewers should reinforce strict adherence to proper reporting guidelines by researchers and article authors to improve reporting quality and reduce spin.

## Supporting information

**S1 Table. List of the trials register that were used to identify eligible studies.**
(DOCX)

**S1 Fig. Overall Consort score for each article.** Articles were de-identified. Y-axis represents the fulfilled CONSORT items per article.
(TIF)

**S2 Fig. Overall Spin Score for each article.** The articles were de-identified. Y-axis represents the numbers of fulfilled Spin checklist items.
(TIF)

**S1 Data.**
(XLSX)

**S2 Data.**
(XLSX)

**S3 Data.**
(XLSX)

## Author Contributions

**Conceptualization:** Effie Ioannidou.

**Data curation:** Murad Shaqman, Khadijeh Al-Abedalla, John Cart Gunsolley, Effie Ioannidou.

**Formal analysis:** Khadijeh Al-Abedalla, Helen Swede, John Cart Gunsolley, Effie Ioannidou.

**Funding acquisition:** Effie Ioannidou.

**Investigation:** Murad Shaqman, Khadijeh Al-Abedalla, Effie Ioannidou.

**Methodology:** Khadijeh Al-Abedalla, Julie Wagner, John Cart Gunsolley, Effie Ioannidou.

**Project administration:** Effie Ioannidou.

**Resources:** Effie Ioannidou.

**Supervision:** Julie Wagner, Helen Swede, John Cart Gunsolley, Effie Ioannidou.

**Visualization:** Khadijeh Al-Abedalla, Effie Ioannidou.

**Writing – original draft:** Murad Shaqman, Khadijeh Al-Abedalla, Effie Ioannidou.

**Writing – review & editing:** Murad Shaqman, Khadijeh Al-Abedalla, Julie Wagner, Helen Swede, John Cart Gunsolley, Effie Ioannidou.

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
