## [Decision Letter · Decision Letter 0]

9 Jan 2020

PONE-D-19-31524

Reporting quality and spin in abstracts of randomized clinical trials of periodontal therapy and cardiovascular disease outcomes

PLOS ONE

Dear Dr Ioannidou,

Thank you for submitting your manuscript to PLOS ONE. After careful consideration, we feel that it has merit but does not fully meet PLOS ONE’s publication criteria as it currently stands. Therefore, we invite you to submit a revised version of the manuscript that addresses the points raised during the review process.

Both reviewers found this to be an interesting paper and worthy of publication. However, they did have a few issues that they would like addressed, in particular, the relationship between spin and propaganda.

We would appreciate receiving your revised manuscript by Feb 23 2020 11:59PM. To enhance the reproducibility of your results, we recommend that if applicable you deposit your laboratory protocols in protocols.io, where a protocol can be assigned its own identifier (DOI) such that it can be cited independently in the future. For instructions see: http://journals.plos.org/plosone/s/submission-guidelines#loc-laboratory-protocols

We look forward to receiving your revised manuscript.

Kind regards,

Dermot Cox

Academic Editor

PLOS ONE

2. Please upload a copy of Figure 1, to which you refer in your text on page 7 and 10. If the figure is no longer to be included as part of the submission please remove all reference to it within the text.

Reviewers' comments:

Reviewer's Responses to Questions

**Comments to the Author**

1. Is the manuscript technically sound, and do the data support the conclusions?

Reviewer #1: Yes

Reviewer #2: Yes

2. Has the statistical analysis been performed appropriately and rigorously? 

Reviewer #1: N/A

Reviewer #2: Yes

3. Have the authors made all data underlying the findings in their manuscript fully available?

Reviewer #1: No

Reviewer #2: Yes

4. Is the manuscript presented in an intelligible fashion and written in standard English?

Reviewer #1: Yes

Reviewer #2: Yes

5. Review Comments to the Author

Reviewer #1: This is an interesting topic, and one that is gathering pace recently – probably belatedly! I have very few comments, mainly to do with phrasing and updating the literature.

1. Introduction

1.1 Page 4: Spin is probably not the same as propaganda, since spin is the biased reporting of factual information in such a way as to mislead the reader, while propaganda can (and usually does) include lies.

1.2 The introduction is well written, but needs a little updating to reflect recent publications. The authors might want to check these :

1. Alharbi F, Almuzian M. The quality of reporting RCT abstracts in four major orthodontics journals for the period 2012-2017. J Orthod. 2019;46(3):225-34.

2. Austin J, Smith C, Natarajan K, Som M, Wayant C, Vassar M. Evaluation of spin within abstracts in obesity randomized clinical trials: A cross-sectional review. Clin Obes. 2019;9(2):e12292.

3. Boutron I, Haneef R, Yavchitz A, Baron G, Novack J, Oransky I, et al. Three randomized controlled trials evaluating the impact of "spin" in health news stories reporting studies of pharmacologic treatments on patients'/caregivers' interpretation of treatment benefit. BMC Med. 2019;17(1):105.

4. Hua F, Sun Q, Zhao T, Chen X, He H. Reporting quality of randomised controlled trial abstracts presented at the SLEEP Annual Meetings: a cross-sectional study. BMJ Open. 2019;9(7):e029270.

5. Roszhart JI, Kumar SS, Allareddy V, Childs CA, Elangovan S. Spin in abstracts of randomized controlled trials in dentistry: A cross-sectional analysis. J Am Dent Assoc. 2019.

2. Methods

Clearly described.

2.1 Page 10 : Cohen's kappa is a statistic, not a test.

2.2 Page 10 : 'non parametric' should be omitted. Statistical procedures are parametric or nonparametric, but data are just data.

3. Results

3.1 The results are interesting, but of course with a small sample they come with wide confidence intervals. I would include these in table 4, as these constitute the focal findings of the study. For instance, claiming efficacy with no consideration of non-significant primary outcome, which is 9 of 14 studies is 64% with a CI of 39% to 84%. So even if the lower figure were the true prevalence, it would still be pretty damning. I don't know if SPSS can be persuaded to do reasonable confidence intervals for small sample sizes (I haven't used it for some years) but the authors should use either Wilson's or Jeffrey's confidence interval. See

Brown, L., Cai, T., DasGupta, A. (2001). Interval estimation for a binomial proportion Statistical Science 16(2), 101 - 117. https://dx.doi.org/10.1214/ss/1009213286

for a discussion of the various formulas for confidence intervals.

3.2 I'd like an explanation of "inadequate extrapolation to larger population" – does this mean overgeneralising the results, or failing to provide adequate confidence intervals? Both, I think, are sins.

3.3 Page 20 : "made hyped statements" hype is deception carried out in publicity. But it can also mean exaggerated claims. I think that "misleading statements designed to exaggerate or falsely claim efficacy" is clearer, and makes explicit what is going in.

3.4 The abbreviations for reporting quality score and spin score make the paper less readable, and reduce the impact. Abbreviations don't save paper, so the authors should confine themselves to ones that are used in everyday language (eg everyone says RCT nowadays) or ones that are accepted acronyms (eg CONSORT)

3.5 The graphs are not helpful because they display the data in no sort of order, leaving the reader unable to form an overall impression. I suggest boxplots with data shown, as in the uploaded picture (produced by jamovi, a free, open-source package that is going to eat SPSS!)

4. Discussion

4.1 "deemed to be poor" = poor.

4.2 "in regard to the spin analysis" = "in the analysis of spin"

4.3 I would prefer "intentionally misleading conclusions" to "hyped conclusions"

Reviewer #2: The take of spin is of course exiting and novel, at least in the dental field. The authors start off with declaring that Spin is just another word for propaganda and then present a list of 20 points that need to be included in an abstract to not be considered as propaganda. In the age of fake news this can easily be misinterpreted as all research paper abstracts failing one of the points should be considered to be propaganda.

The results section needs to be more focused on Spin and its criteria. The abstract is more spin oriented than the results section, which is a bit ironic.

In the discussion section the authors need to add a section in which they discuss the concept of spin – to use the same verdict for insinuating that non-significant data are significant as for putting some spotlight on a secondary outcome that turned out to be interesting than the authors thought when designing the study is something that should be discussed. It needs to be made extremely clear (also in the abstract) that an abstract failing one of the 20 point (in table 4) is not the same thing as fraudulent research. The paper and the research its presenting could still be excellent.

Finally the authors need to compare their data to similar Spin analysis in other medical fields. Right now that’s just one sentence, it need to be elaborated.

6. PLOS authors have the option to publish the peer review history of their article (what does this mean?). If published, this will include your full peer review and any attached files.

Reviewer #1: Yes: Ronán M Conroy, DSc

Reviewer #2: No

---

## [Author Response · Author response to Decision Letter 0]

18 Feb 2020

Reviewer #1: This is an interesting topic, and one that is gathering pace recently – probably belatedly! I have very few comments, mainly to do with phrasing and updating the literature.

We thank the review for the kind words.

1. Introduction

1.1 Page 4: Spin is probably not the same as propaganda, since spin is the biased reporting of factual information in such a way as to mislead the reader, while propaganda can (and usually does) include lies.

The reviewer raises a valid concern. Therefore, the term ”propaganda” was removed from the text. 

1.2 The introduction is well written, but needs a little updating to reflect recent publications. The authors might want to check these:

1. Alharbi F, Almuzian M. The quality of reporting RCT abstracts in four major orthodontics journals for the period 2012-2017. J Orthod. 2019;46(3):225-34.

2. Austin J, Smith C, Natarajan K, Som M, Wayant C, Vassar M. Evaluation of spin within abstracts in obesity randomized clinical trials: A cross-sectional review. Clin Obes. 2019;9(2):e12292.

3. Boutron I, Haneef R, Yavchitz A, Baron G, Novack J, Oransky I, et al. Three randomized controlled trials evaluating the impact of "spin" in health news stories reporting studies of pharmacologic treatments on patients'/caregivers' interpretation of treatment benefit. BMC Med. 2019;17(1):105.

4. Hua F, Sun Q, Zhao T, Chen X, He H. Reporting quality of randomised controlled trial abstracts presented at the SLEEP Annual Meetings: a cross-sectional study. BMJ Open. 2019;9(7):e029270.

5. Roszhart JI, Kumar SS, Allareddy V, Childs CA, Elangovan S. Spin in abstracts of randomized controlled trials in dentistry: A cross-sectional analysis. J Am Dent Assoc. 2019.

We appreciate highlighting these references to us. The papers as recommended by the reviewer were cited appropriately. 

2. Methods

Clearly described.

2.1 Page 10: Cohen's kappa is a statistic, not a test. 

We revised the manuscript accordingly and the word “test” was changed into “statistic”.

2.2 Page 10: 'non parametric' should be omitted. Statistical procedures are parametric or nonparametric, but data are just data.

The phrase “non parametric” was removed in page 10.

3. Results

3.1 The results are interesting, but of course with a small sample they come with wide confidence intervals. I would include these in table 4, as these constitute the focal findings of the study. For instance, claiming efficacy with no consideration of non-significant primary outcome, which is 9 of 14 studies is 64% with a CI of 39% to 84%. So even if the lower figure were the true prevalence, it would still be pretty damning. I don't know if SPSS can be persuaded to do reasonable confidence intervals for small sample sizes (I haven't used it for some years) but the authors should use either Wilson's or Jeffrey's confidence interval. See

Brown, L., Cai, T., DasGupta, A. (2001). Interval estimation for a binomial proportion Statistical Science 16(2), 101 - 117. https://dx.doi.org/10.1214/ss/1009213286

for a discussion of the various formulas for confidence intervals.

Although the reviewer raises a good point, the limited sample size prevented us from using inferential statistics to avoid misleading conclusions.

3.2 I'd like an explanation of "inadequate extrapolation to larger population" – does this mean over-generalizing the results, or failing to provide adequate confidence intervals? Both, I think, are sins.

Boutron et al. define this item as “over generalizing the results”. Therefore, when we applied the checklist, we followed their definition.

3.3 Page 20: "made hyped statements" hype is deception carried out in publicity. But it can also mean exaggerated claims. I think that "misleading statements designed to exaggerate or falsely claim efficacy" is clearer, and makes explicit what is going in.

We agreed with the recommendation. Therefore, the item title was changed into “misleading statements designed to exaggerate or falsely claim efficacy”.

3.4 The abbreviations for reporting quality score and spin score make the paper less readable, and reduce the impact. Abbreviations don't save paper, so the authors should confine themselves to ones that are used in everyday language (e.g. everyone says RCT nowadays) or ones that are accepted acronyms (e.g. CONSORT)

We agreed with the recommendation and we revised the manuscript accordingly.

3.5 The graphs are not helpful because they display the data in no sort of order, leaving the reader unable to form an overall impression. I suggest boxplots with data shown, as in the uploaded picture (produced by jamovi, a free, open-source package that is going to eat SPSS!)

We agreed with the recommendation and we revised the supplement figures into box plots.

4. Discussion

4.1 "deemed to be poor" = poor.

4.2 "in regard to the spin analysis" = "in the analysis of spin"

4.3 I would prefer "intentionally misleading conclusions" to "hyped conclusions"

We agreed with the recommendation and these rephrasing suggestions were adopted. 

Reviewer #2: 

Comment 1: The take of spin is of course exiting and novel, at least in the dental field. The authors start off with declaring that Spin is just another word for propaganda and then present a list of 20 points that need to be included in an abstract to not be considered as propaganda. In the age of fake news this can easily be misinterpreted as all research paper abstracts failing one of the points should be considered to be propaganda.

The reviewer raises a valid concern. We strived to be sensitive to this aspect both during the preparation of the manuscript and during this review process. Accordingly, and to avoid sending out a wrong message, the term “propaganda” was removed. Also, some changes in the wording of the abstract were made to reflect our findings, as much as an abstract allows, with least possible misinterpretation. Moreover, the distinction between poor reporting and poor research design/ execution, and between presence of spin and fraud was highlighted and emphasized further in our discussion section. 

Comment 2: The results section needs to be more focused on Spin and its criteria. The abstract is more spin oriented than the results section, which is a bit ironic.

The criteria of spin according to Boutron et al are outlined/elaborated in table 4. In addition, our findings regarding each spin criterion were presented. Our reporting of results highlights these findings. We thank the reviewer for the comment and we made an effort to balance CONSORT and Spin in in both abstract and result sections. 

Comment 3: In the discussion section the authors need to add a section in which they discuss the concept of spin – to use the same verdict for insinuating that non-significant data are significant as for putting some spotlight on a secondary outcome that turned out to be interesting than the authors thought when designing the study is something that should be discussed. It needs to be made extremely clear (also in the abstract) that an abstract failing one of the 20 point (in table 4) is not the same thing as fraudulent research. The paper and the research its presenting could still be excellent.

The reviewer raises a valid and concern and we refer him to our response to his first comment/ recommendation. Specifically, changes were made in the abstract and we included a dedicated paragraph in the discussion section to highlight this point 

Comment 4: Finally, the authors need to compare their data to similar Spin analysis in other medical fields. Right now, that’s just one sentence, it needs to be elaborated.

We agree with the recommendation and we elaborated on our discussion of relevant medical literature. We emphasized similarities to our findings but also an interesting finding of discordance between abstracts and the full text of a sample of studied RCT’s. 

In summary, we hope that the revisions are satisfactory and the manuscript is now accepted for publication. 

Sincerely

Effie Ioannidou

UCONN Health

Ioannidou@uchc.edu

---

## [Decision Letter · Decision Letter 1]

11 Mar 2020

Reporting quality and spin in abstracts of randomized clinical trials of periodontal therapy and cardiovascular disease outcomes

PONE-D-19-31524R1

Dear Dr. Ioannidou,

We are pleased to inform you that your manuscript has been judged scientifically suitable for publication and will be formally accepted for publication once it complies with all outstanding technical requirements.

With kind regards,

Dermot Cox

Academic Editor

PLOS ONE

Additional Editor Comments (optional):

Reviewers' comments:

Reviewer's Responses to Questions

**Comments to the Author**

1. If the authors have adequately addressed your comments raised in a previous round of review and you feel that this manuscript is now acceptable for publication, you may indicate that here to bypass the “Comments to the Author” section, enter your conflict of interest statement in the “Confidential to Editor” section, and submit your "Accept" recommendation.

Reviewer #1: All comments have been addressed

Reviewer #2: All comments have been addressed

2. Is the manuscript technically sound, and do the data support the conclusions?

Reviewer #1: Yes

Reviewer #2: Yes

3. Has the statistical analysis been performed appropriately and rigorously? 

Reviewer #1: Yes

Reviewer #2: Yes

4. Have the authors made all data underlying the findings in their manuscript fully available?

Reviewer #1: No

Reviewer #2: Yes

5. Is the manuscript presented in an intelligible fashion and written in standard English?

Reviewer #1: Yes

Reviewer #2: Yes

6. Review Comments to the Author

Reviewer #1: The authors have made a good job of the revision. I tend to agree with them that for small numbers like this, and given the lack of clarity around what population could be generalised to, confidence intervals are not appropriate.

Reviewer #2: Well done! Accepted as it stands. The paper has actually made an impression on me and I'm looking forward to citing it.

7. PLOS authors have the option to publish the peer review history of their article (what does this mean?). If published, this will include your full peer review and any attached files.

Reviewer #1: Yes: Ronán M Conroy

Reviewer #2: No

---

## [Editor Report · Acceptance letter]

23 Mar 2020

PONE-D-19-31524R1 

Reporting quality and spin in abstracts of randomized clinical trials of periodontal therapy and cardiovascular disease outcomes 

Dear Dr. Ioannidou:

I am pleased to inform you that your manuscript has been deemed suitable for publication in PLOS ONE. Congratulations! Your manuscript is now with our production department. 

With kind regards,

on behalf of

Dr. Dermot Cox 

Academic Editor

PLOS ONE